# CRISPR-Cas9 genome and long non-coding RNAs as a novel diagnostic index for prostate cancer therapy via liposomal-coated compounds

**Mai O. Kadry** *, **Rehab M. Abdel-Megeed**

Therapeutic Chemistry Department, National Research Center, Al Bhoouth Street, Cairo, Egypt

* maiosman666@yahoo.com

**Data Availability Statement:** The underlying data can be accessed using the following DOI: 10.6084/m9.figshare.25416790.

## Abstract

CRISPR/Cas9 is a recently discovered genomic editing technique that altered scientist's sight in studying genes function. Cas9 is controlled via guide (g) RNAs, which match the DNA targeted in cleavage to modify the respective gene. The development in prostate cancer (PC) modeling directed not only to novel resources for recognizing the signaling pathways overriding prostate cell carcinoma, but it has also created a vast reservoir for complementary tools to examine therapies counteracting this type of cancer. Various cultured somatic rat models for prostate cancer have been developed that nearly mimic human prostate cancer. Nano-medicine can passively target cancer cells via increasing bioavailability and conjugation via specific legend, contributing to reduced systemic side-effects and increased efficacy. This article highlights liposomal loaded Nano-medicine as a potential treatment for prostate cancer and clarifies the CRISPR/Cas9 variation accompanied with prostate cancer. PC is induced experimentally in western rat model via ethinyl estradiol for 4 weeks and SC. dose of 3, 2'- dimethyl-4-aminobiphenyl estradiol (DAE) (50mg/kg) followed by treatment via targeted liposomal-coated compounds such as liposomal dexamethasone (DXM), liposomal doxorubicin (DOX) and liposomal Turmeric (TUR) (3mg/kg IP) for four weeks in a comparative study to their non-targeted analogue dexamethasone, doxorubicin and Turmeric. 3, 2'- dimethyl-4-aminobiphenylestradiol elicit prostate cancer in western rats within 5 months. Simultaneous supplementations with these liposomal compounds influence on prostate cancer; tumor markers were investigated via prostate-specific antigen (PSA), Nitric oxide (NOX) and CRISPR/Cas9 gene editing. Several long non-coding RNAs were reported to be deregulated in prostate cell carcinoma, including *MALAT1*. On the other hand, gene expression of apoptotic biomarkers focal adhesion kinase (AKT-1), phosphatidylinistol kinase (PI3K) and glycogen synthase kinase-3 (GSK-3) was also investigated and further confirming these results via histopathological examination. Liposomal loaded dexamethasone; doxorubicin and Turmeric can be considered as promising therapeutic agents for prostate cancer via modulating CRISPR/Cas9 gene editing and long non coding gene *MALAT1*.

**Funding:** Mai O Kadry received funding from National research Centre, project grants, awards number 13020251.

**Competing interests:** The authors have declared that no competing interests exist

## Introduction

Prostate cell carcinoma still remains undefeated in the history of mankind with yearly mortality rate over millions worldwide. Consequently, novel techniques for struggling cancer are developing via various researchers all over the world. PC is defined as the uncontrolled growth of malignant cells that begins in stromal cells that line the prostate. Prostate cancer five-year relative survival rate is nearly 90%.

Growing evidence declares that PC is extensively widespread male cancer and contributes to the second contributing cause of mortality worldwide [1]. Early prognosis, paves the way for radiotherapy, prostatectomy and convenient therapy. In the late stage of bone metastases, no therapeutic option is accessible, stressing the requirement for new therapeutic options. In primary and metastatic cancers, tumor cells closely interact with different cell types and the extracellular matrix constituting the stromal compartment. Many stages of cancer metastasis, carcinogenesis, and dissemination are known to be significantly influenced by tumor-associated inflammation [2]. Prostate carcinogenesis has been linked to various inflammatory cell types [3]. Neoplastic cells can stimulate different stromal cell types, and stimulated stromal cells can release additional growth factors that later promote the invasion and proliferation of cancer cells [4]. Interfering with tumor-associated inflammation offers a promising, yet understudied, method to treat cancer since pro-inflammatory stromal cells have a power to promote tumor growth [5]. Dexamethasone (DEX), a highly promising anti-inflammatory therapeutic drug, and other glucocorticoids (GC) [6] are frequently used chemotherapy for soothing effects in PC treatment. However, it is yet unknown whether GC really does reduce tumor-related inflammation to provide an additional therapeutic effect. To provide such a particular anti-tumor action, it has been hypothesized that substantial tumor concentrations of GC are required [7]. A probable explanation for their limited usage in cancer therapy is that such tissue concentrations can only be reached by substantial and subsequent GC doses, which comes with the well-known range of harmful GC-related side effects [6,8]. Glucocorticoids have the ability to suppress the immune system and make these cells resistant to apoptosis, this may aid in the spread of tumors.

Traditionally, cancer remedy involves radiology, surgery and chemotherapeutic drugs as DEX, cisplatin and DOX. Nevertheless, DEX and DOX mostly face the obstacle of limited uptake. To improve the stability and to co- deliver these drugs slowly into the target system, methoxypoly (ethylene glycol)-poly(ε-caprolactone) (MPEG-PCL) micelles overloaded with dexamethasone and DOX (Dex-Dox/MPEG-PCL) was previously investigated, that enhanced antitumor action in PC subjects [9].

Through the latest few eras, targeted liposomal medicine has developed a progressive remedial regimen. Many cancers are categorized by restricted lymphatic drainage and restricted vasculature. The improved permeability and retention phenomena are the ability of long-circulating liposome to accumulate and gradually concentrate in the tumor tissue [10]. Liposome as an effective delivery for anti-inflammatory medications to the supportive tumor microenvironment is justified by the prevalence of tumor associated macrophages and their effective phagocytizing capabilities. Liposome limit the exposure of healthy cells to that encapsulated medicine, which can greatly reduce the therapeutic agent's toxicity. These characteristics subsequently justify the use of long-circulating liposomal drugs in cancer treatment, with liposomal DOX serving as an example [11]. Liposomal Nano-medicines, anti-tumor effects against prostate cancer were investigated in vivo in a number of relevant preclinical models [12] and in clinical investigations [13]. The therapeutic approach of long-circulating liposomal GC administration is being developed for immunological disorders and has proven a promising impact against arthritis and cancer model [14] and preclinical models of cancer [7,15].

Unfortunately, liposomal Glucocorticoids were not yet used in clinical prostate cancer investigations. Herein, the efficacy of liposomal DEX, DOX and TUR versus their free liposomal analogue in rat prostate cancer model was investigated. Likewise, the pharmacokinetic profiling and toxicological impact of Lip-DOX, DEX and TUR was investigated [16]. TUR limited breast cancer metastasis to lung tissue was contributed to reducingMMP-9, survivin, GSK-3, PI3K, AKT, NF-κB, and cyclooxygenase [17] and STAT1 stimulation, thus limiting cancer progress and promoting apoptosis inA549 and H1299 cells [18]. TUR enhanced the cell proliferation inhibitory activity and augmented CIS apoptotic action in prostate adenocarcinoma resistant cells [19].

Many long non-coding RNAs are active and contribute to the development of cancer through regulatory mechanisms. Long non-coding RNAs, including PCGEM, PCAT-1, MALAT, and the prostate cancer gene 3 (PCA3), have recently been discovered to be deregulated in PC [20,21], describing the PCA3 gene's extreme over expression in prostatic cancers. It has been suggested that PCA3 controls PRUNE2 levels by generating a double-stranded RNA [22,23]. PRUNE2 is thought to be involved in the control of neoplastic transformation. In human malignancies, such as those of the prostate, breast, pancreatic, colon, and liver, MALAT1 was over expressed. This suggests that urinary MALAT1 may be a promising diagnostic biomarker for PC [24,25].

Protein linked with CRISPR 9 (Cas9) is an enzyme that recognizes and cleaves particular DNA strands that are complementary to CRISPR sequences using the CRISPR sequences as a guide. The CRISPR-Cas9 technique, which may be used to modify the genes of organisms [25], is based on Cas9 enzymes and CRISPR sequences. The production of biotechnological products, the treatment of diseases, and fundamental biological research are just a few of the many uses for this editing process [26]. By using CRISPR-Cas9 technology to remove the ER gene from the mouse genome, researchers have discovered that it controls the growth of the ventral prostate and functions as a tumor suppressor [25–27]. The clustered regularly interspaced short palindrome repeats (CRISPR) and CRISPR-associated (Cas) 9 systems, which can efficiently and persistently decrease gene expression in vitro and in vivo, were not used in these earlier investigations. The manner that scientists explore genes and their activities in mammalian systems has significantly changed as a result of the recent discovery of the genome editing system CRISPR/Cas9.It is developed from the bacterial immune system CRISPR/Cas and Cas9 is guided by guide (g) RNAs that correspond to the targeted DNA cleavage to change the relevant gene. The CRISPR/Cas9 system was used to knockout the expression of PC-associated miRNAs such as miR-205, miR-221, miR-455-3p, miR-222, miR-224, miR-505, miR-23b, miR-30c, miR-1225-5p, and miR-663a in LNCaP cells in order to quickly classify the differential functions of miRNAs that were identified through miRNA expression profiling in PC [25]. It was declared that when comparing PC tissues to benign tissues, 37 miRNAs were down regulated and 14 were up regulated. Another previous work used miRNA microarray analysis to examine the differential expression of miRNAs in PC, and discovered 11 elevated miRNAs and 17 reduced miRNAs. The current study aims to examine the effectiveness of liposomal loaded dexamethasone, doxorubicin, and turmeric in treating CRISPR-Cas9, long non-coding RNAs, as well as Lnc-RNAs generated in rat models of prostate cancer in addition to some related molecular pathways including AKT, PI3K and GSK-3.

## Materials and methods

### Chemicals

Liposomal doxorubicin, liposomal dexamethasone, liposomal Turmeric, doxorubicin, dexamethasone and Turmeric were purchased from Sigma-Aldrich Co (St. Louis, MO, USA) and

Avanti (Germany) companies. RT-PCR kits of PI3K, AKT and GSK3 were provided from Qiagen USA.ELISA kits for prostate specific antigen (PSA) were obtained from R & D systems (MN, USA) and long non coding RNA *(MALAT1)* kits were provided from Qiagen (USA).All other chemicals were of the highest analytical grade.

### Animals and treatments

Sixty four male Western Albino rats weighing 170–190 g (6–8 weeks) old, from the animal house of National Research Center were used in this study. Animals were kept at standardized conditions (22 ± 5˚C, 55 ± 5% humidity, and 12 h light/dark cycle). They were allowed free access to water and pelleted standard chow diet. All procedures relating to animal care and treatments strictly adhered to the ethical procedures (13020251–1) and policies approved by Animal Care and Use Committee of National Research Center, and US National Institute of Health.

### Experimental design

1 week post acclimatization, animals were randomly divided into eight groups (8 rats each) and were divided according to the following schedule:

**Group1:** Animals received saline as control group.

**Group 2:** prostate cancer induced experimentally in rats via a diet containing 0.75 ppm of ethinyl estradiol for 3 weeks and a basal diet for 2 weeks alternately 10 times, and 2 days after each change to basal diet, a single SC. injection of 50 mg/kg body weight of 3,2'- dimethyl-4-aminobiphenyl estradiol (DAE) respectively as prostate cancer model [8]. Rats were kept for 8 months till the incidence of prostate cancer then were treated as follows:

**Groups 3:** DAE group was cured with liposomal carried doxorubicin in a dose of (3 mg/kg BW) IP for 1 month post prostate cancer induction via DAE [21,22].

**Groups 4:** DAE group was cured via doxorubicin in a dose of (5 mg/kg BW) IP for 1 month post prostate cancer induction [22].

**Groups 5:** DAE group was cured via liposomal carried dexamethasone in a dose of (3 mg/kg BW) IP for 1 month [22].

**Groups 6:** DAE group was cured via dexamethasone in a dose of (5 mg/kg BW) IP for 1 month post prostate cancer induction [22].

**Groups 7:** DAE group was cured via liposomal carried Turmeric in a dose of (3 mg/kg BW) IP for 1 month [22].

**Groups 8:** DAE group was cured via Turmeric in a dose of (5 mg/kg BW) IP for 1 month [22].

## 1. Blood sampling and tissue preparation

To alleviate animal suffering, non-invasive techniques was utilized, created a safe and comfortable home for the animal, applying anesthesia during the procedure, and employing analgesic regimens (Isofurane) to relieve discomfort throughout the recovery. At the final experimental period, rats were, slightly anesthetized via carbon dioxide and blood samples were collected from the sublingual vein. Sera was separated by centrifugation at 4000 rpm for 10 min and was kept at − 80˚C for subsequent estimation of biochemical and molecular analysis.

Animals were then sacrificed via cervical dislocation and prostate tissue was carefully separated, and then divided into portions. The first portion was homogenized in 4 volumes of phosphate buffer, pH 7.4, using Teflon homogenizer (Glass-Col homogenizer, Terre Haute, USA). An aliquot of this homogenate (20% w/v) was centrifuged at 4000 rpm at 4 $^O$C for 15 min and the supernatant was used for ELISA determination. The second portion of the prostate was used for RT-PCR, long non coding RNA estimation. The remaining portion was kept in 10% formaldehyde, and then embedded in paraffin for subsequent histopathological examination.

## 2. Measured biochemical parameters

### Prostate cancer tumor markers:

ELISA kits were used to measure the prostate specific antigen (PSA, P-AKT, PI3K &GSK-3) activity (R &D systems MN, USA). The quantitative sandwich enzyme immunoassay approach was estimated by the tests. The microplate has a pre-applied coating of a particular antibody. The immobilized antibody bound PSA, P-AKT, PI3K &GSK-3 after the standards and samples were pipette into the wells, and the absorbance was measured at 450 nm [23].

### Prostate Nitric oxide:

Nitric oxide was estimated by spectrophotometer via Randox Company kits according to manufactures instructions by monitoring the reddish-purple azo dye at 540 nm.

### mRNA gene expression of prostate AKT, PI3K and GSK-3:

RT-PCR was performed to detect the target gene expression analysis utilizing particular forward and reverse primers for AKT, PI3K, and GSK-3. Promega, Madison, Wisconsin's SV total RNA isolation system was used to first extract the total RNA from the prostate tissue samples. The isolated RNA was then reverse-transcribed into cDNA and amplified by PCR using an RT-PCR kit (Stratagene, USA). A final volume of 50 μL was used for the reactions (25 μL SYBR Green Mix (2x), 0.5 μL cDNA, 2 μL primer pair mix (5 pmol/μL each primer), and 22.5 μL water). The PCR reaction will look like this: 50°C for 2 minutes, 95°C for 10 minutes, 45° to 60°C for 30 seconds, 72°C for 30 seconds, and 72°C for 10 minutes [24]. Forward and Reverse primer sequences of selected genes are listed in Table 1.

### Long non coding gene analysis for *MALAT1*

The mRNA MALAT was detected via (qRT-PCR) in prostate. RNA was extracted via RT2 miscript kits (Qiagen, USA) specific for Lnc genes extraction followed by RT2-SYBR green kits for quantitative RT-PCR determination of MALAT (Lnc gene) [25,26].

### CRISPR/Cas9 guide RNA design

Target sequences of miR-122-5p for CRISPR interference were designed. Taking miR-205 as an example (Gene ID406988: The gene sequence 5′-AAAGATCCTCAGACAATCCATGTGCTT CTCTTGTCCTTCATTCCACCGGAGTATACCCAACCAGATTTCAGTGGAGTGAAGTTCAGGA GGCATGGAGC-3′ was retrieved and downloaded from Gene Bank and the sgRNA was designed by Zhang lab using Target Finder (version 2014, and DNA 2.0 gRNA. A total of six optimal target sequences for each miRNA were chosen and five random sequences were utilized as controls corresponding to androgenic genes [27].

**Table 1. Forward and reverse primers sequence for GSK-3, PI3K, CRISPR and AKT.**

| Gene | primer |
|---|---|
| GSK-3 | 5'-GGAACTCCAACAAGGGAGCA- 3'<br>5' -TTCGGGGTCGGAAGACCTTA-3' |
| PI3K | 5' -CCA GAC CCT CAC ACT CAG ATCA-3'<br>5' -TCC GCT TGG TGG TTT GCT A-3' |
| CRISPR | 5'AAAGATCCTCAGACAATCCATGTGCTTCTCTTGTCCTTCATTCCACCGGAGTA-3'5'<br>TACCCAACCAGATTTCAGTGGAGTGAAGTTCAGGAGGCATGGAGC-3' |
| AKT | 5' -CAT GAA GAG AAG ACA CTG ACC ATG GAAA-3'<br>5' -TGG ATA GAG GCT AAG TGT AGA CAC G-3' |
| B-Actin | 5' -TGG AGT CTA CTG GCG TCT T-3'<br>5' -TGT CAT ATT TCT CGT GGT TCA-3 |

**Histopathological examination:** The prostate tissue was embedded in paraffin blocks, and then sliced into 5 μm in thickness. After hematoxylin–eosin (HE) staining, the slides were observed under optical microscope [28].

## Statistical analysis

Results were expressed as mean ± standard error mean (SEM) and was carried out by one-way analysis of variance (ANOVA) test, SPSS program, at $p < 0.05$.

# Results

## Modulation of prostate cancer biomarker

Ethinyl estradiol revealed a significant elevation in prostate cancer biomarker PSA. Meanwhile treatment with liposomal loaded dexamethasone, doxorubicin and Turmeric as well as their non-targeted analogue significantly modulated PSA with liposomal Turmeric revealing the most significant impact (Fig 1) revealing prostate cancer induction.

## Modulation of angiogenesis biomarker

Ethinyl estradiol revealed a significant elevation in angiogenesis biomarker NOX. Meanwhile treatment with liposomal loaded dexamethasone, doxorubicin and Turmeric as well as their non-targeted analogue significantly modulated NOX with liposomal Turmeric revealing the most significant impact (Fig 2). Revealing increased angiogenesis to prostate cancer cell and oxidative stress.

## Modulation of cell survival biomarkers

Herein, ethinyl estradiol elucidated a significant up regulation in PI3K, AKT as well as a significant down regulation in GSK3 gene and protein expression (Table 2). Meanwhile treatment with liposomal dexamethasone; doxorubicin and Turmeric as well as their non-targeted analogue significantly modulated these deviated genes with the superiority of Turmeric and liposomal Turmeric in this field (Figs 3–5).

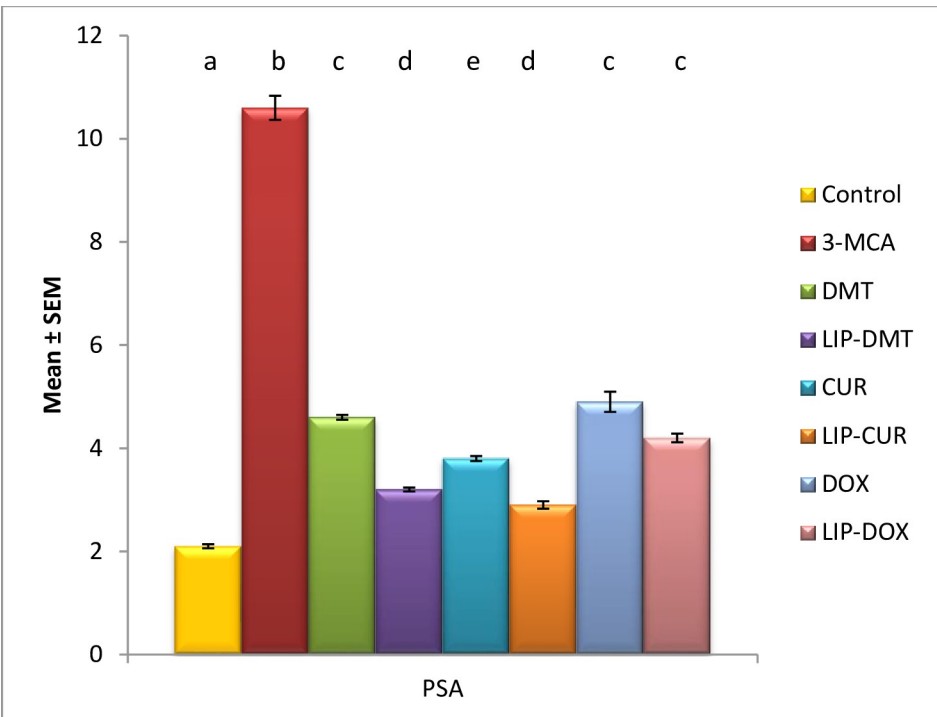

**Fig 1. Effect of liposomal loaded dexamethasone, Turmeric and doxorubicin and their non-liposomal analogue on serum PSA post 3- estradiol induced prostate cancer.** Data are expressed as mean ± S.E.M (n = 8). P ≤ 0.05 value is considered significant. Groups having the same letter are not significantly different from each other, while those having different letters are significantly different from each other.

## Modulation of CRISPR genome editing

Herein, ethinyl estradiol deduced a significant elevation in CRISPR genome editing. Meanwhile treatment with liposomal dexamethasone, doxorubicin and Turmeric as well as their non-targeted analogue significantly modulated this deviated gene with liposomal doxorubicin revealing the most significant impact (Fig 6). Reflecting the beneficial role of GRISPR gene editing as a prognostic and diagnostic tool for prostate cancer progression.

## Modulation of long non coding RNA (*Lnc MALAT*)

Fig 7, declared that ethinyl estradiol deduced a significant up regulation in long non coding gene *MALAT-1*. Meanwhile, treatment with liposomal dexamethasone; doxorubicin and Turmeric as well as their non-targeted analogue significantly modulated this non coding gene with liposomal Turmeric revealing the most significant impact. Sparkling *Lnc-RNA (MALAT-1)* as a prospective prostate cancer biomarker in addition to, heatmap diagram representing all investigated genes and their correlations (Fig 8).

## Histopathological examination

3- estradiol group showed massive number of desquamated lining epithelium in the acinar lumen associated with thickening in the interacinar stroma. There was cystic dilatation in the acinar lumen with thin lining epithelium. Meanwhile, group of rats experimentally inducted and treated by liposomal dexamethasone showed mild hyperplasia with polyps formation in the lining acinar epithelial cells. Further, group of experimentally inducted rats and treated by

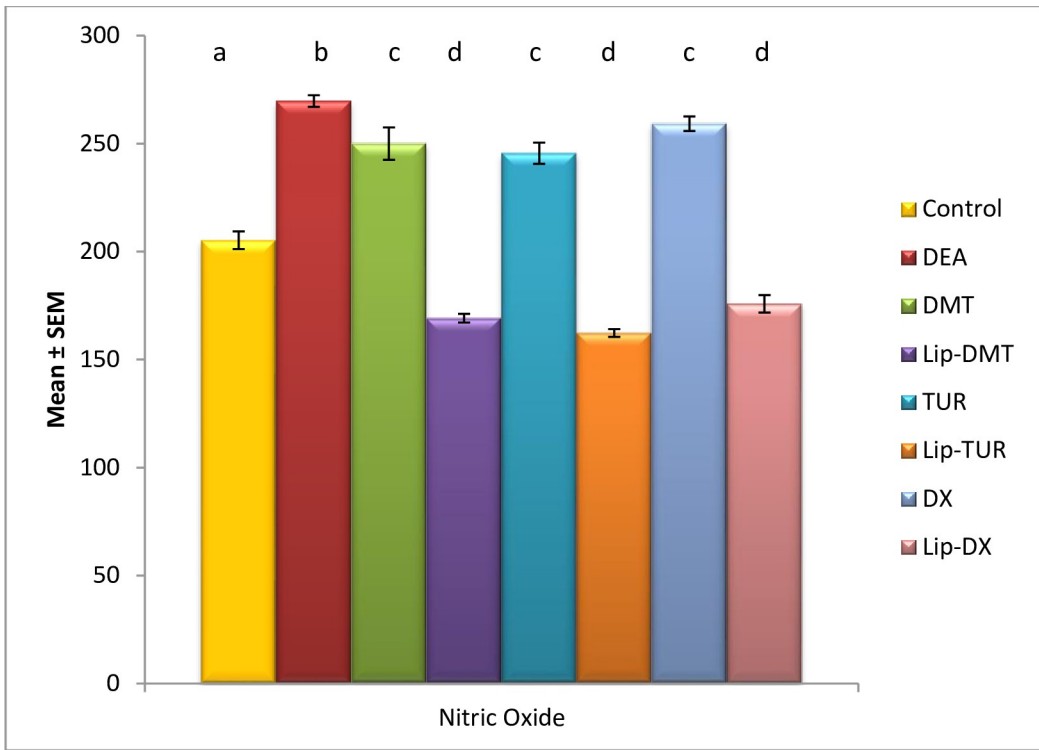

**Fig 2. Effect of liposomal loaded dexamethasone, Turmeric and doxorubicin and their non-liposomal analogue on serum Nitric oxide post 3- estradiol induced prostate cancer.** Data are expressed as mean ± S.E.M (n = 8). P ≤ 0.05 value is considered significant. Groups having the same letter are not significantly different from each other, while those having different letters are significantly different from each other.

liposomal turmeric, the acinar lumen showed cystic dilatation associated with eosinophilic secretion in the lumen. In addition, group of experimentally inducted rats and treated by liposomal doxorubicin: There was no histopathological alteration (Fig 9).

**Table 2. Impact of liposomal loaded dexamethasone, Turmeric and doxorubicin and their non-liposomal analogue on GSK-3, p-AKT-1 and PIK-3 protein expression post 3- estradiol induced prostate cancer using ELISA technique.**

| Groups/parameters | GSK-3 (ng/ml) | P-AKT(ng/ml) | PIK-3(ng/ml) |
|---|---|---|---|
| Control | 5.1±0.53 [a] | 2.8±0.12 [a] | 3.54±0.3[a] |
| 3-MCA | 2.4±0.32 [b] | 6.2±2.3 [b] | 8.02±0.87[b] |
| DMT | 3.76±0.54[c] | 3.76±1.08[c] | 3.06±0.3[a] |
| LIP-DMT | 2.9±0.23[b] | 2.93±0.98[a] | 4.2 ±0.13[c] |
| CUR | 4.41±0.61[d] | 4.53±0.71[d] | 3.2±0.43[a] |
| LIP-CUR | 2.74±0.11[b] | 3.45±0.54[c] | 3.7±0.24[a] |
| DOX | 4.63±0.24[d] | 4.82±1.03[d] | 4.01±0.2[c] |
| LIP-DOX | 3.52±0.15[c] | 4.04±1.23[d] | 4.23±0.3[c] |

Data are expressed as mean ± SEM (n = 8). p ≤ 0.05 value is considered significant. Groups having the same letter are not significantly.

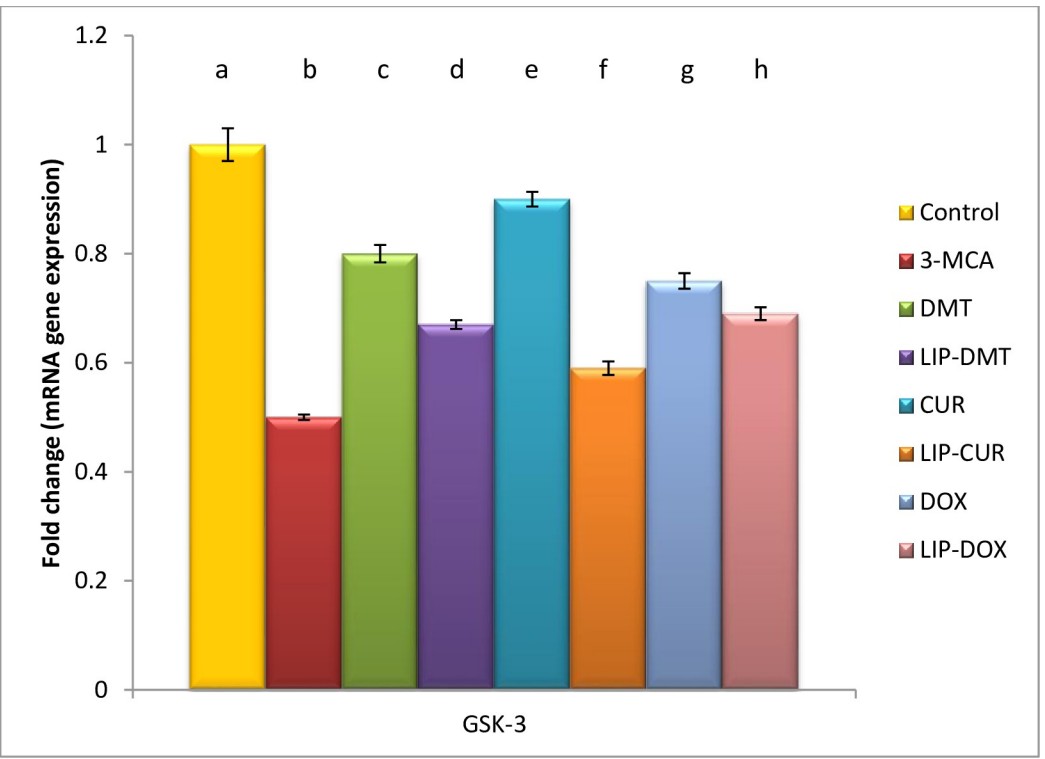

**Fig 3. Effect of liposomal loaded dexamethasone, Turmeric and doxorubicin and their non-liposomal analogue on GSK-3 gene expression post 3- estradiol induced prostate cancer.** B-actin was used as reference gene. Data are expressed as mean ± S.E.M (n = 8). P ≤ 0.05 value is considered significant. Groups having the same letter are not significantly different from each other, while those having different letters are significantly different from each other.

## Discussion

PC is converting normal cells abnormally into malignant cells. Certain PCs develop steadily and are symptomless for years while others develop faster and are more aggressive. Biopsy is the only tool that can predict the degree of cancer. Nevertheless, there are numerous novel bio-markers and genomic analysis accessible which can help in determining if the cancer will be aggressive or not. PC is the most widely spread type of cell carcinoma and the second contributing cause of cancer deaths in males [16].

Herein, the efficacy of liposomal loaded DEX, DOX and TUR versus their free non-liposomal analogue in rat prostate cancer model was investigated.

Herein, ethinyl estradiol deduced a significant alteration in GSK-3, PI3K and AKT gene and protein expression. In the meantime, liposomal dexamethasone; doxorubicin and Turmeric administration significantly modulated these altered genes with liposomal Turmeric revealing the profound significant impact. This accord with the postulate that Akt-1 contributes to enhanced cell survival, proliferation and glucose metabolism [29,30]. Promoting Akt pathway rely on the duration and frequency of PIP3availability at the plasma membrane. PIP3 is responsible for AKT phosphorylation via PI3K [31]. PI3K convert PIP2 into PIP3. This occurs post the activation of a tyrosine kinase receptor via attaching of IGF-1 to IGF-R. Akt, PDK1 and PDK2 must join PIP3 for Akt activation via phosphorylation [32,33]. Several downstream signaling proteins are involved in a cascade that is initiated by p-Akt (activated phosphorylated Akt) [34]. Whereas p-Akt is undetectable in healthy prostate tissue, a high level of

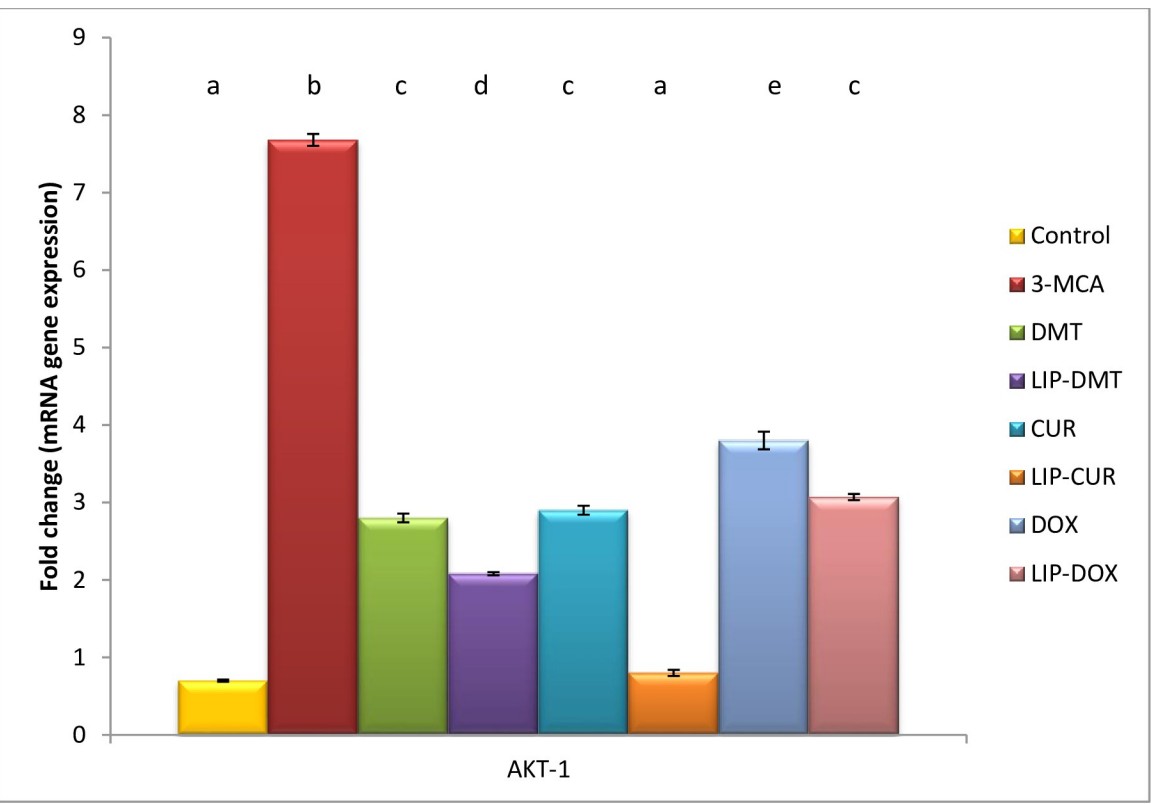

**Fig 4. Effect of liposomal loaded dexamethasone, Turmeric and doxorubicin and their non-liposomal analogue on AKT-1 gene expression post 3- estradiol induced prostate cancer.** B-actin was used as reference gene. Data are expressed as mean ± S.E.M (n = 8). P ≤ 0.05 value is considered significant. Groups having the same letter are not significantly different from each other, while those having different letters are significantly different from each other.

p-Akt is associated with a bad prognosis for prostate cancer [35]. GSK3 (glycogen synthase kinase-3), which prevents the up-regulation of cell proliferation and improved cyclin D1 breakdown, is inhibited by activated Akt. On the other hand, limiting GSK-3 significantly prevents the breakdown of free catenin in PC [36]. Normally, catenin functions as a bridge alongside the cytoskeleton in cadherin motifs and cell-to-cell contacts. Furthermore, elevated catenin can join and stabilize cyclin D1 mRNA, resulting in elevated expression that accelerates the transition between the G1- and S-phases. Cells enter the early G1 (growth) phase [24] when Cyclin D1 is expressed. A characteristic of prostate neoplastic transformation is increased cyclin D1 production, which promotes cell-cycle progression even in androgen- or serum-deprived cells [37]. The cyclin/Cdk4 (cyclin-dependent kinase 4) complex is elevated as a result of the accumulation of cyclin D1, which promotes the progression of the G1/S phase of the cell cycle, and this elevation is caused by a secret process [24].

Herein, ethinyl estradiol deduced a significant elevation in CRISPR gene editing of androgenic hormones that are obviously deviated in prostate cancer. Meanwhile treatment with liposomal dexamethasone, doxorubicin and Turmeric significantly modulated this deviated gene with liposomal doxorubicin revealing the most significant impact. This is consistent with the theory that CRISPR-associated protein 9 (Cas9) is an enzyme that recognizes and cuts particular DNA strands that correspond to CRISPR sequences by using CRISPR sequences as a guide [38]. CRISPR-Cas9 technology, may be used to modify the genes of organisms, is built on CRISPR sequences gathered with Cas9 enzymes. This editing procedure is extensively used in

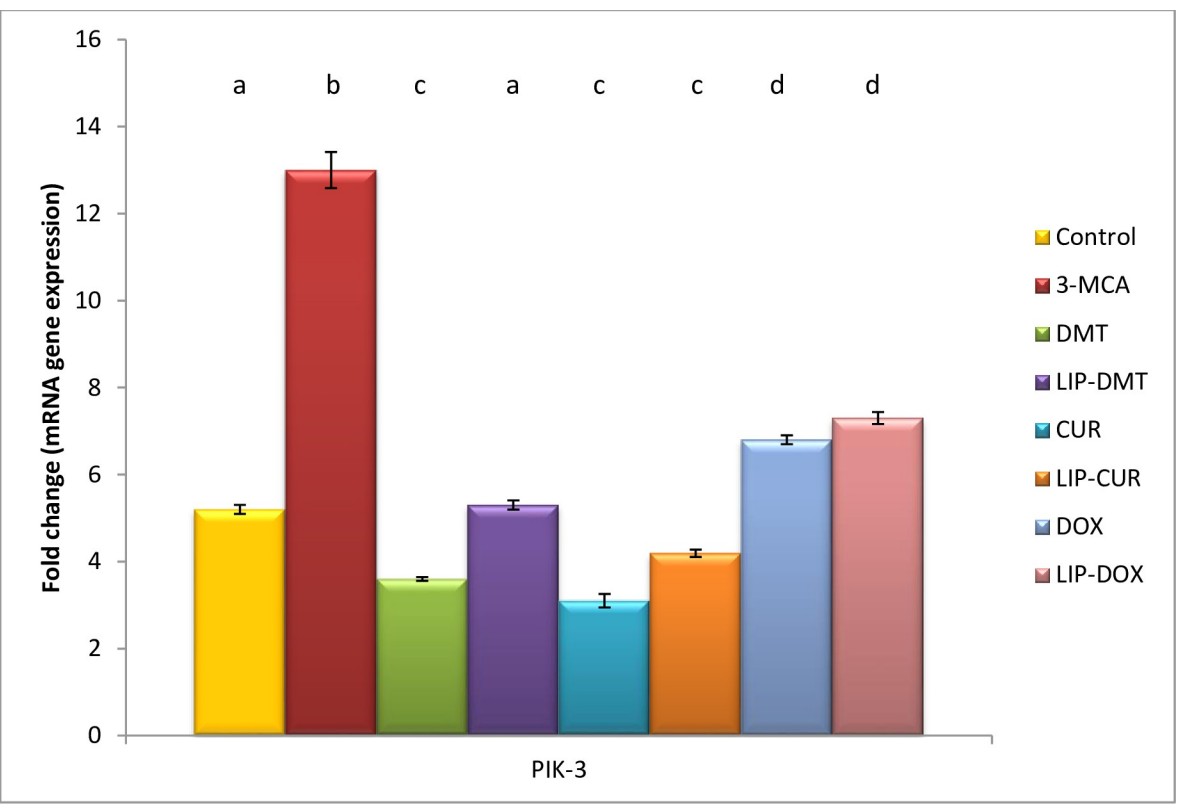

**Fig 5. Effect of liposomal loaded dexamethasone, Turmeric and doxorubicin and their non-liposomal analogue on PIK-3 gene expression post 3-estradiol induced prostate cancer.** B-actin was used as reference gene. Data are expressed as mean ± S.E.M (n = 8). P ≤ 0.05 value is considered significant. Groups having the same letter are not significantly different from each other, while those having different letters are significantly different from each other.

the production of biotechnological products, basic biological research, and treatment of diseases [39]. By removing the ER gene using the CRISPR-Cas9 technology, it was discovered that ER controls the growth of the ventral prostate and functions as a tumor suppressor gene [40]. Moreover, PC features distinct miRNA expression profiles that serve as the foundation for the functional analysis of miRNA in PC [41]. 11 miRNAs were up-regulated and 17 miRNAs were down-regulated in PC [41,42]. The method that scientists examine genes and their activities in mammalian systems has been significantly changed by the breakthrough genome editing technology CRISPR [43]. It is an offshoot of the CRISPR/Cas9 bacterial immune system, and Cas9 that is guided by guide (g) RNAs corresponding to the DNA targeted for cleavage to change the relevant gene. miR-222, miR-455-3p,miR-221, miR-205, miR-224, miR-663a, miR-505, miR-23b, miR-30c, and miR-1225-5p were among the PC-related miRNAs that were knocked down using the CRISPR/Cas9 system in LNCaP cells [42,44].

Ethinyl estradiol was shown to significantly up-regulate the long non-coding gene MALAT-1 in this study. Liposomal dexamethasone, doxorubicin, and turmeric therapy effectively reduced these aberrant genes, with liposomal turmeric having the most influence. There has been evidence that a significant number of long non-coding RNAs are functional and contribute to the development of cancer through regulatory mechanisms. In a novel development, the PCA3 prostate cancer gene, PCAT-1, MALAT-1 PCGEM, and several other long non-coding RNAs were investigated for potential deregulation in PC. [45] revealed an over expression

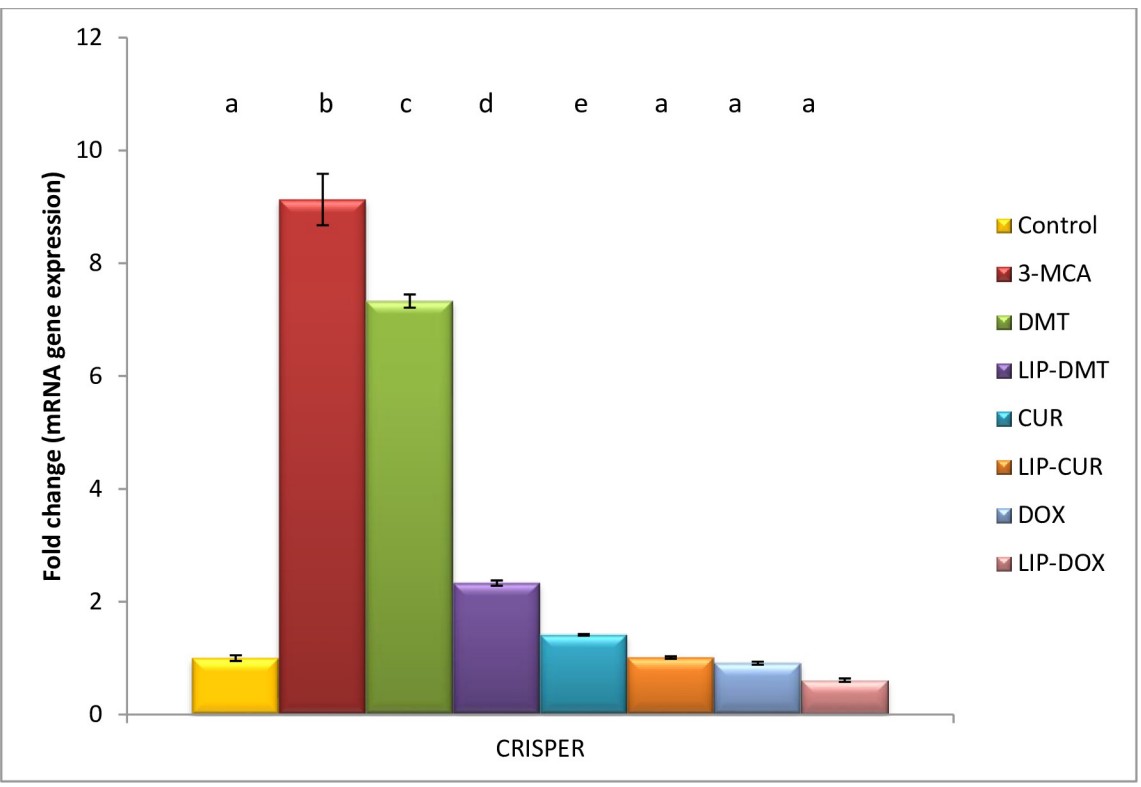

**Fig 6. Effect of liposomal loaded dexamethasone, Turmeric and doxorubicin and their non-liposomal analogue on CRISPR gene expression post 3- estradiol induced prostate cancer.** B-actin was used as reference gene. Data are expressed as mean ± S.E.M (n = 8). $P \leq 0.05$ value is considered significant. Groups having the same letter are not significantly different from each other, while those having different letters are significantly different from each other.

in the PCA3 gene. It has been hypothesized that PCA3 controls PRUNE2 levels by producing double-stranded RNA [46,47].

Radiology, surgery, and chemotherapy medications like DOX, cisplatin, and dexamethasone are all included in traditional cancer care. Yet, DEM and DOX usually face the difficulty of lower intake. Methoxy poly (ethylene glycol)-poly(-caprolactone) micelles loaded with DEM and DOX (Dex-Dox/MPEG-PCL) were investigated to improve the stability and sustained co-delivery of these medicines into the target system, which improved antitumor responses in PC model [9].

A novel approach in drug delivery system that is still developing is targeted liposomal medication. Numerous cancers have inadequate lymphatic drainage and weak vasculature. The increased permeability and retention (EPR) impact [48] is the ability of specially created long-circulating liposomes to extravagate and gradually accumulate in tumor tissue following intravenous delivery. The use of liposomes for the effective delivery of anti-inflammatory medications to the supporting tumor microenvironment is justified by the presence of tumor associated macrophages and their effective phagocytizing capabilities. Liposomes often limit the exposure of healthy tissues to the medicine that is encapsulated, which can greatly lessen the therapeutic agent's toxicity. As a result, these properties support the use of long-circulating liposome medications in the management of cancer, as demonstrated by liposomal doxorubicin (also known as Doxil/Caelyx) [11]. Liposomal cytotoxic medicines' anti-tumor effects against prostate cancer were observed in vivo in a number of relevant preclinical models [12]

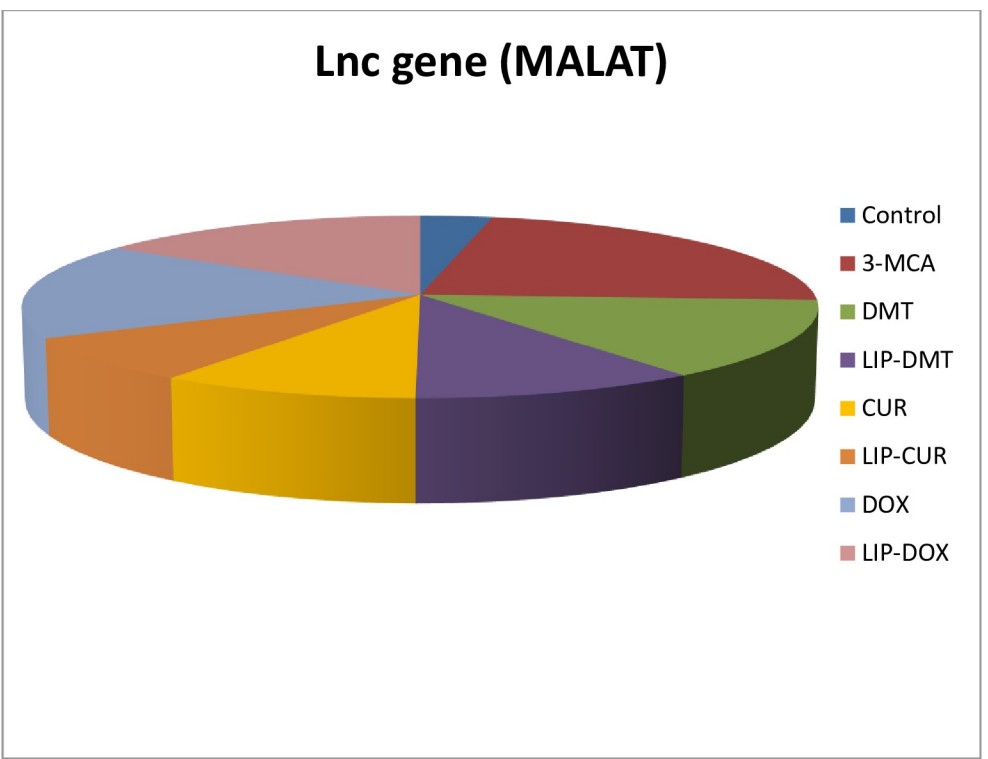

**Fig 7. Impact of liposomal dexamethasone, Turmeric and doxorubicin and their non-liposomal analogue on the percent of long non coding gene *MALAT* post 3- estradiol induced prostate cancer.** GAPDH was used as reference gene. Data are expressed as percentage (n = 8). P $\leq$ 0.05 value is considered significant.

and in clinical investigations [13]. A recent treatment technique for autoimmune and cancer illnesses, as well as for preclinical cancer models [7,15,16], is the introduction of GC-loaded liposomes.

Owing to decreased expression of GSK-3, MMP-9, NF-B, and COX17, TUR revealed reduced lung metastasis of breast cancer. In H1299 and A549 cells, TUR has been demonstrated to suppress STAT1 activity, which inhibits cancer development and triggers apoptosis [18]. In DDP-resistant prostate cancer cells, TUR increased the cell proliferation inhibitory activity and encouraged the apoptotic activity of cisplatin (DDP) [19].

Since it was designed to be safe, anti-inflammatory and antioxidant agent TUR has a wide range of therapeutic and pharmacologic benefits. In addition to combining its effectiveness as a therapeutic and preventive agent against cancer, TUR has opened the door for its future use as an anti-inflammatory agent to combat ROS and inflammation. Enzymes, cytokines, growth factors and their receptors, as well as proteins that control apoptosis and cell proliferation, are just a few of the molecules that TUR targets [49]. It is difficult to design TUR-based drugs, and the process becomes more difficult due to their systemic elimination, poor absorption, rapid metabolism and restricted bioavailability [19]. TUR can lessen tumor bulk and cancer progression as well as prevent cancer relapse [50,51]. Yet, TUR extremely low bioavailability is also a part of this success story of turmeric. Other strategies are being developed to address this issue, such as the creation of targeted liposomal loaded TUR. TUR affects also the immune system, lowering the body's capacity to tolerate cancer [18]. Moreover, TUR has been demonstrated to influence the cell's micro-RNAs and other epigenetic modifiers, slowing the progression of

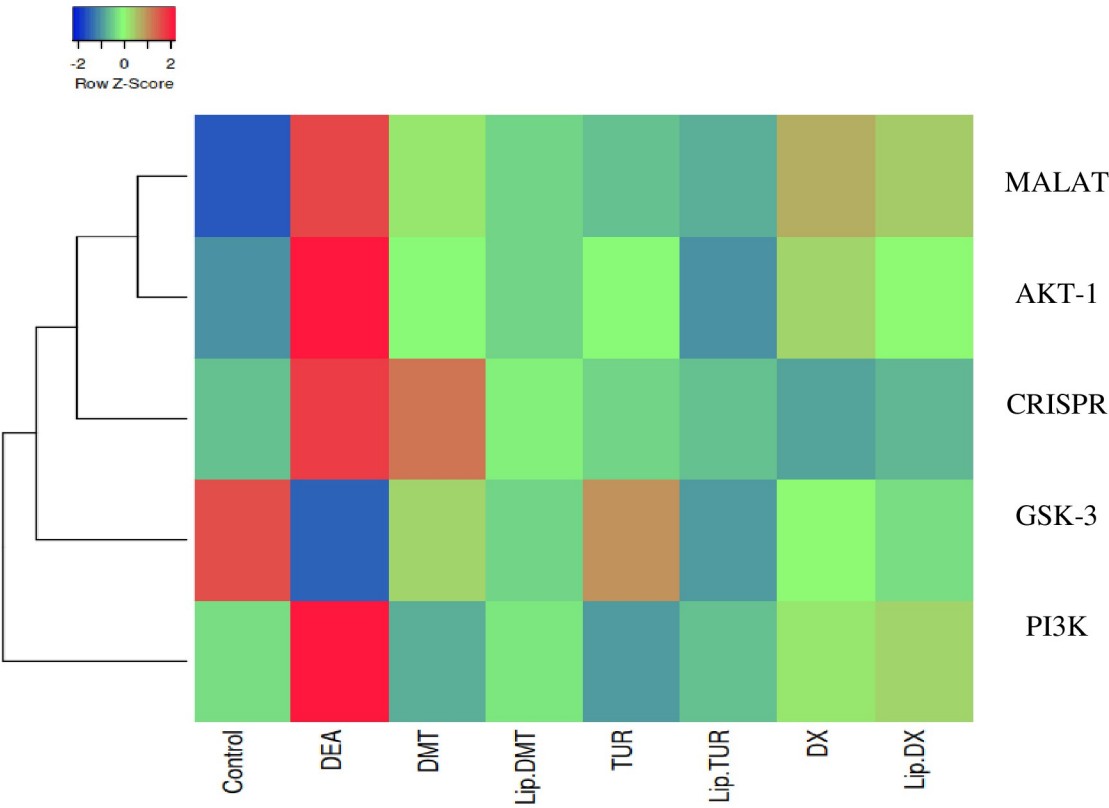

**Fig 8. Heatmap representing different gene expression; blue reflects low score meanwhile red reflects high score.**

cancer [52,53]. One of the most popular methods for delivering small compounds, peptides, nucleic acid, and proteins is the use of Nano-liposomes [54]. The phospholipids phosphatidyl choline, phosphatidyl glycerol and phosphatidyl ethanolamine, as well as cholesterol, spontaneously self-organize to produce Nano-liposomes. Amazingly, many of the lipids used to produce liposomes are important components of naturally occurring bilayers [55,56]. The dual polar and non-polar regions of liposomal bilayers, which enable nonpolar medications to emerge in the lipid bilayer or polar drugs to be contained in the central water core, are their main distinguishing feature. Biochemistry, medicine, biology, cosmetics, and food production all use Nano-liposomes [57]. Increased retention time, minimal toxicity, the capacity to change size and surface, and biocompatibility are all beneficial characteristics of nano-liposomes [58]. Diverse strategies like liposomes, Nano-gels and microspheres have been utilized to bypass the obstacle of weak absorption, rapid metabolism, fast systemic elimination and other clinical restrictions of turmeric. Lip-TUR possesses high growth inhibitory and pro-apoptotic impact [59]. Solid dispersion of tocopheryl PEG with turmeric was formulated to increase oral bioavailability, solubility dissolution rate and cell permeability of TUR. Sustained release, high therapeutic index and cellular uptake and good solubility were observed post Nano-gel formulation of TUR. Micro-emulsion of TUR can cross BBB and prevent the development of glioblastoma [59].

Doxorubicin impact on cancer cells starts via passive diffusion across the membrane phospholipid bilayer of malignant cells to the cytoplasm, there it is converted into a semiquinone and release ROS, generating free radical and oxidative stress. In the cytoplasm, DOX pass in the mitochondria initiating DNA damage and energetic stress. Thus, the mitochondria discharge

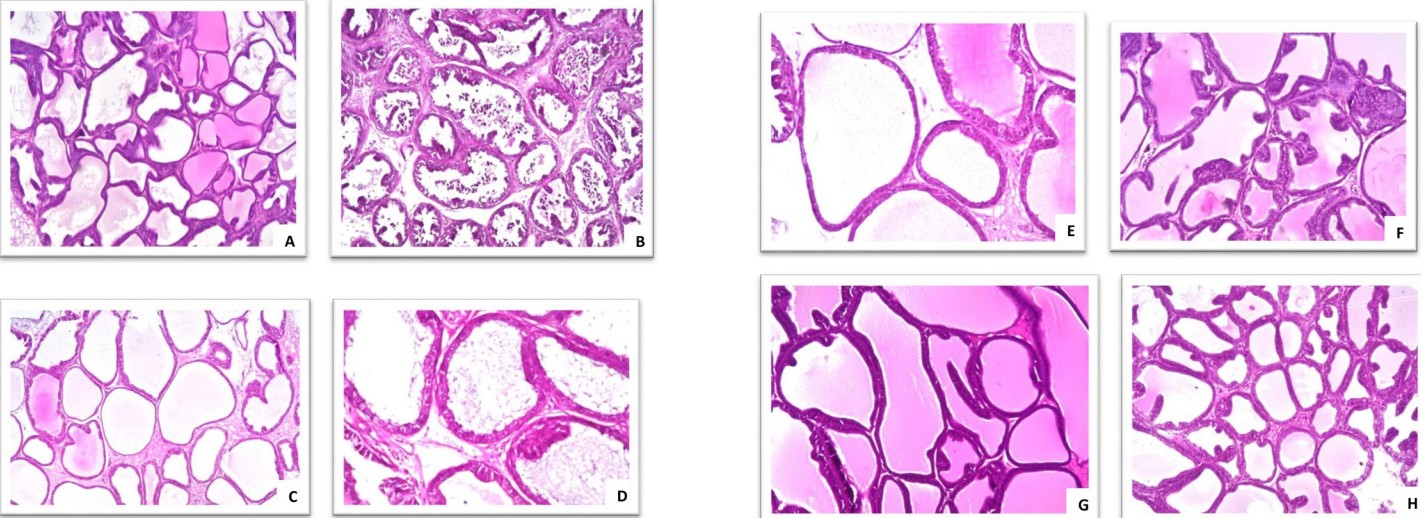

**Fig 9.** Histopathological findings: [**A**] group of rats kept as control: There was no histopathological alteration and the normal histological structure of the acini with cuboidal tall lining epithelium with eosinophilic secretion in the lumen.[**B**] Group of experimentally inducted prostate cancer by administration of 3- estradiol: Massive number of desquamated lining epithelium were detected in the acinar lumen associated with thickening in the interacinarstroma. There was cystic dilatation in the acinar lumen with thin lining epithelium. [**C**] Group of rats experimentally inducted and treated by dexamethasone: Mild hyperplasia with polyps formation were detected in the lining acinar epithelial cells. [**D**] Group of experimentally inducted rats and treated by curcumin: The acinar lumen showed cystic dilatation associated with eosinophilic secretion in the lumen. [**E**] Group of experimentally inducted rats and treated by Doxorubicin: There was no histopathological alteration. [F] Group of rats treated with Liposomal dexamethasone revealed no histopathological alteration [G] Group of rats treated with Liposomal curcumin revealed normal histopathological architecture [H] Group of rats treated with Liposomal Doxorubicin revealed no histopathological alteration.

the cytochrome *C* protein, promoting the caspase signaling cascade leading to apoptosis. From the cytosol, DOX translocate into the nucleus where it intercalates between double-stranded DNA helices and inhibits the enzymes topoisomerases I and II. The resulting damage to DNA leads to free radical generation, alkylation, and activation of the p53 pathway. Nevertheless, the therapeutic potential of DOX is limited by its cardio-toxicity. To overwhelm this problem, the lip-DOX formulation was established to limit DOX-related cardio-toxicity while preserving its antitumor effectiveness. Lip-DOX encapsulates DOX within a phospholipids bilayer that is coated with methoxy polyethylene glycol. The PEGylation protects the liposomes from recognition by the mononuclear phagocyte system (MPS) and allows a longer circulation time in the bloodstream while reducing the exposure of free DOX circulating in the plasma [60].

## Conclusion

In particular, CRISPR-Cas9 gene editing and long non-coding RNAs approaches are in great demand for the assessment of prognostic and diagnostic tool for cancer progression, therapeutic efficacy and safety margin of targeted and non-targeted drugs that might be useful as drugs for treatment of prostate cancer. Further, we are in pressing need to such new technology "Targeted Nano-medicine" to target specific organ associated with cancer without harming normal tissues, so they are within the scope of Nano-biotechnology. By combining nanotechnology with the newly developing technology long non-coding RNAs CRISPR genome editing in this research, we could achieve the new effective and productive drugs against prostate cancer.

## Acknowledgments

Is directed to Prof DR Adel Bekeer, Faculty of Dentistry Cairo University, for histopathological examination.

## Author Contributions

**Data curation:** Mai O. Kadry.

**Formal analysis:** Mai O. Kadry, Rehab M. Abdel-Megeed.

**Funding acquisition:** Mai O. Kadry.

**Methodology:** Mai O. Kadry, Rehab M. Abdel-Megeed.

**Project administration:** Mai O. Kadry.

**Supervision:** Mai O. Kadry.

**Writing – review & editing:** Mai O. Kadry.

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
