## [Decision Letter · Decision Letter 0]

4 Jan 2024

PONE-D-23-28438CRISPR-Cas9 genome as a novel prognostic tool for Prostate cancer: long non-coding RNAs the predictive diagnostic index of targeted liposomal-coated compounds in prostate cancer therapyPLOS ONE

Dear Dr. Kadry,

Thank you for submitting your manuscript to PLOS ONE. After careful consideration, we feel that it has merit but does not fully meet PLOS ONE’s publication criteria as it currently stands. Therefore, we invite you to submit a revised version of the manuscript that addresses the points raised during the review process.

We look forward to receiving your revised manuscript.

Kind regards,

Laila Adel Ziko

Academic Editor

PLOS ONE

Journal Requirements:

Did you know that depositing data in a repository is associated with up to a 25% citation advantage (https://doi.org/10.1371/journal.pone.0230416)? If you’ve not already done so, consider depositing your raw data in a repository to ensure your work is read, appreciated and cited by the largest possible audience. You’ll also earn an Accessible Data icon on your published paper if you deposit your data in any participating repository (https://plos.org/open-science/open-data/#accessible-data).

The name of the colleague or the details of the professional service that edited your manuscript.A copy of your manuscript showing your changes by either highlighting them or using track changes (uploaded as a *supporting information* file).A clean copy of the edited manuscript (uploaded as the new *manuscript* file).

4. To comply with PLOS ONE submissions requirements, in your Methods section, please provide additional information regarding the experiments involving animals and ensure you have included details on (1) methods of sacrifice, (2) methods of anesthesia and/or analgesia, and (3) efforts to alleviate suffering.

"Mai O Kadry received funding from National research Centre, project grants, awards number 13020251"

6. Please note that in order to use the direct billing option the corresponding author must be affiliated with the chosen institute. Please either amend your manuscript to change the affiliation or corresponding author, or email us at plosone@plos.org with a request to remove this option.

7. Please include a separate caption for each figure in your manuscript.

**Additional Editor Comments:**

Please address the reviewers’ comments thoroughly and send it back to us for the second revision.

Reviewers' comments:

Reviewer's Responses to Questions

**Comments to the Author**

1. Is the manuscript technically sound, and do the data support the conclusions?

Reviewer #1: Yes

Reviewer #2: Yes

2. Has the statistical analysis been performed appropriately and rigorously? 

Reviewer #1: Yes

Reviewer #2: Yes

3. Have the authors made all data underlying the findings in their manuscript fully available?

Reviewer #1: Yes

Reviewer #2: Yes

4. Is the manuscript presented in an intelligible fashion and written in standard English?

Reviewer #1: Yes

Reviewer #2: Yes

5. Review Comments to the Author

Reviewer #1: It's an interesting study. The author used the liposomal loaded DXM,DOX,and TUR to treat the DAE-induced PC, and shows a reduced dosage for the drugs and good efficacy on the biomarkers of PC, including PSA, NOX, as well as the MALAT1 and apoptosis signaling, such as AKT,PI3K and GSK-3. However, it will be better if they author could complete the last Figure HE staining with all control groups (without liposomal loaded DXM,DOX and TUR) and the protein expression of ATK.PI3K and GSK-3.

Reviewer #2: 1. What’s the current techniques to defeat prostate cell carcinoma? What’s the five-year survival rate for prostate cell carcinoma?

2. What’s the obstacles for the clinical application of liposomal glucocorticoids in prostate cancer?

3. What’s the individual effect of CRISPR/Cs9 on PC-associated miRNAs? Which PC-associated miRNA is affected mostly by CRISPR/Cas9?

4. In the method of mRNA gene expression of prostate AKT, PI3K and GSK-3, the volume should be �l rather than L.

5. In Fig.1, what’s the reason for no difference between DOX and LIP-DOX?

6. In Fig.1, 2, 3, 4, 5 & 6, the authors should indicate the P-value number rather than the different letters.

7. In Fig.7, the authors should select other long-noncoding genes as positive and negative controls.

6. PLOS authors have the option to publish the peer review history of their article (what does this mean?). If published, this will include your full peer review and any attached files.

Reviewer #1: **Yes: **Ping Han It's an interesting study. The author used the liposomal loaded DXM,DOX,and TUR to treat the DAE-induced PC, and shows a reduced dosage for the drugs and good efficacy on the biomarkers of PC, including PSA, NOX, as well as the MALAT1 and apoptosis signaling, such as AKT,PI3K and GSK-3. However, it will be better if they author could complete the last Figure HE staining with all control groups (without liposomal loaded DXM,DOX and TUR) and the protein expression of ATK.PI3K and GSK-3.

Reviewer #2: No

---

## [Author Response · Author response to Decision Letter 0]

5 Feb 2024

Editorial comments

ETHICS NUMBER WAS REMOVED and kept only in the materialand methods. 

Tables was inserted inthe manuscript.

funder financially supported the work but had No role in study design ,data collection and analysis ,decision to publish or preparation of the manuscript. please I encountered a problem inmy labtop so kindly add this tothe cover letter.

Please ensure that your manuscript meets PLOS ONE's style requirements, including those for file naming Adjusted as required

We suggest you thoroughly copyedit your manuscript for language usage, spelling, and grammar Adjusted as required

To comply with PLOS ONE submissions requirements, in your Methods section, please provide additional information regarding the experiments involving animals and ensure you have included details on (1) methods of sacrifice, (2) methods of anesthesia and/or analgesia, and (3) efforts to alleviate suffering. Adjusted as required

Please state what role the funders took in the study. NRC is financially supported the current work

Please include a separate caption for each figure in your manuscript. Adjusted as required

Please include captions for your Supporting Information files at the end of your manuscript, and update any in-text citations to match accordingly. Please see our Supporting Information guidelines for more information Adjusted as required

Reviewers comments

What’s the current techniques to defeat prostate cell carcinoma? Different strategies such as liposomes, solid dispersion, complex, emulsion, micelles, nanogels and microspheres have been employed to overcome poor absorption and other limitations of curcumin [1].

DOX’s mechanism of action on cancer cells begins with its passive diffusion through the phospholipid bilayer membrane of malignant cells into the cytoplasm, where DOX is converted into a semiquinone and generates reactive oxygen species (ROS), causing free radical formation and oxidative stress. In the cytosol, DOX enters the mitochondria causing DNA damage and energetic stress. As a result, the mitochondria release the cytochrome C protein, triggering the caspase cascade leading to cell death. From the cytosol, DOX translocates into the nucleus where it intercalates between double-stranded DNA helices and inhibits the enzymes topoisomerases I and II. The resulting damage to DNA leads to free radical generation, alkylation, and activation of the p53 pathway

However, the potential therapeutic benefits of DOX are limited by the risk of cardiotoxicity, which has been evidently related to its lifetime cumulative dose. To overcome this hurdle, the liposomal DOX (L-DOX) formulation was developed in order to reduce DOX-associated cardiotoxicity while preserving its antitumor efficacy.10 The L-DOX formulation encapsulates DOX within a phospholipid bilayer that is coated with methoxypolyethylene glycol. The PEGylation protects the liposomes from recognition by the mononuclear phagocyte system (MPS) and allows a longer circulation time in the bloodstream while reducing the exposure of free DOX circulating in the plasma.

1-Feng T, Wei Y, Lee RJ, Zhao L. Liposomal curcumin and its application in cancer. International journal of nanomedicine. 2017 Aug 21:6027-44.

2- Franco YL, Vaidya TR, Ait-Oudhia S. Anticancer and cardio-protective effects of liposomal doxorubicin in the treatment of breast cancer. Breast Cancer (Dove Med Press). 2018 Sep 11;10:131-141. doi: 10.2147/BCTT.S170239. PMID: 30237735; PMCID: PMC6138971.

What’s the five-year survival rate for prostate cell carcinoma? The 5-year relative survival rate for most people with local or regional prostate cancer is nearly 100%. For people diagnosed with prostate cancer that has spread to other parts of the body, the 5-year relative survival rate is 90%.

What’s the obstacles for the clinical application of liposomal glucocorticoids in prostate cancer? Glucocorticoids have been used widely in conjunction with other treatment for patients with cancer because they have potent proapoptotic properties in lymphoid cells, can reduce nausea, and alleviate acute toxic effects in healthy tissue. However, glucocorticoids are used in a supportive-care role, even though to our knowledge no prospective clinical studies have assessed the effect of these steroids on the growth of solid tumours. Data from preclinical and, to some extent, clinical studies, suggest that glucocorticoids induce treatment resistance in solid tumours, including prostate cancer. Research has focussed on disseminated cells that have been shed by the tumour: the potential of glucocorticoids to render these cells resistant to apoptosis--and to downregulate the immune response--might contribute to tumour metastasis. Here, we review the benefits of glucocorticoids and their negative effects, such as induction of resistance in tumour cells and concomitant induction of apoptosis in immune cells, with particular emphasis on prostate cancer.

Reference:

Herr, I., & Pfitzenmaier, J. (2006). Glucocorticoid use in prostate cancer and other solid tumours: implications for effectiveness of cytotoxic treatment and metastases. The Lancet. Oncology, 7(5), 425–430. https://doi.org/10.1016/S1470-2045(06)70694-5

What’s the individual effect of CRISPR/Cs9 on PC-associated miRNAs? Which PC-associated miRNA is affected mostly by CRISPR/Cas9?

 Prostate cancer (PCa) has a distinctive miRNA expression profile, which has been the basis for the functional study of miRNA in PCa (1). Porkka et al (1) demonstrated that 37 miRNAs were downregulated and 14 upregulated in PCa tissues compared with benign tissues. A previous study obtained differential expression data of miRNAs in PCa determined by miRNA microarray analyses and reported that 11 miRNAs were upregulated and 17 miRNAs were downregulated in PCa (2). However, these previous gain-of-function studies did not include loss-of-function analyses using miRNA knockout and applying the clustered regularly interspaced short palindromic repeats (CRISPR) and CRISPR associated (Cas) 9 system, which can effectively, specifically and stably suppress gene expression in vitro and in vivo (3,4). CRISPR/Cas9 is a recently discovered genome editing system, which has markedly changed the way that researchers study genes and their functions in mammalian systems (4,5). It is derived from the CRISPR/Cas bacterial-acquired immune system and Cas9 is directed by guide (g)RNAs, which match the DNA targeted in cleavage to modify the respective gene (3,5,6).

References:

1-Porkka KP, Pfeiffer MJ, Waltering KK, Vessella RL, Tammela TL and Visakorpi T: MicroRNA expression profiling in prostate cancer. Cancer Res 167: 6130-6135, 2007

2- He HC, Han ZD, Dai QS, Ling XH, Fu X, Lin ZY, Deng YH, Qin GQ, Cai C, Chen JH, et al: Global analysis of the differentially expressed miRNAs of prostate cancer in Chinese patients. BMC Genomics 14: 757, 2013.

3-Cho SW, Kim S, Kim JM and Kim JS: Targeted genome engineering in human cells with the Cas9 RNA guided endonuclease. Nat Biotechnol 31: 230-232, 2013. 

4- Cong L, Ran FA, Cox D, Lin S, Barretto R, Habib N, Hsu PD, Wu X, Jiang W, Marraffini LA and Zhang F: Multiplex genome engineering using CRISPR/Cas systems. Science 339: 819-823, 2013 

5- Mali P, Yang L, Esvelt KM, Aach J, Guell M, DiCarlo JE, Norville JE and Church GM: RNA guided human genome engineering via Cas9. Science 339: 823-826, 2013

6- Jiang, F. N., Liang, Y. X., Wei, W., Zou, C. Y., Chen, G. X., Wan, Y. P., ... & Zhong, W. D. (2020). Functional classification of prostate cancer associated miRNAs through CRISPR/Cas9 mediated gene knockout. Molecular Medicine Reports, 22(5), 3777-3784.

l rather than L.�4. In the method of mRNA gene expression of prostate AKT, PI3K and GSK-3, the volume should be corrected

5. In Fig.1, what’s the reason for no difference between DOX and LIP-DOX? 

In Fig.1, 2, 3, 4, 5 & 6, the authors should indicate the P-value number rather than the different letters.

 Added to figure caption

It will be better if they author could complete the last Figure HE staining with all control groups (without liposomal loaded DXM,DOX and TUR) and the protein expression of ATK.PI3K and GSK-3. Added as required

---

## [Editor Report · Decision Letter 1]

8 Feb 2024

PONE-D-23-28438R1CRISPR-Cas9 genome as a novel prognostic tool for Prostate cancer: long non-coding RNAs the predictive diagnostic index of targeted liposomal-coated compounds in prostate cancer therapyPLOS ONE

Dear Dr. Kadry,

Thank you for submitting your manuscript to PLOS ONE. After careful consideration, we feel that it has merit but does not fully meet PLOS ONE’s publication criteria as it currently stands. Therefore, we invite you to submit a revised version of the manuscript that addresses the points raised during the review process.

The reviewers' comments were sufficiently addressed, however the title needs to be grammatically proofread and shortened preferably. Also, there are several grammatical typos throughout the manuscript with words without spaces between them, this needs to be corrected before it can be further accepted.

We look forward to receiving your revised manuscript.

Kind regards,

Laila Adel Ziko

Academic Editor

PLOS ONE

Journal Requirements:

Additional Editor Comments:

The reviewers' comments were sufficiently addressed, however the title needs to be grammatically proofread and shortened preferably. Also, there are several grammatical typos throughout the manuscript with words without spaces between them, this sneedes to be corrected before it can be further accepted.

---

## [Author Response · Author response to Decision Letter 1]

15 Mar 2024

DOI LINK for Data availability was supplied 

Title was adjusted as required

Typo writing and sticky words were adjusted as required

cover letter the role of funder was mentioned

---

## [Editor Report · Decision Letter 2]

1 Apr 2024

CRISPR-Cas9 genome and long non-coding RNAs as a novel diagnostic index for Prostate cancer therapy via liposomal-coated compounds

PONE-D-23-28438R2

Dear Dr. Kadry,

We’re pleased to inform you that your manuscript has been judged scientifically suitable for publication and will be formally accepted for publication once it meets all outstanding technical requirements.

Kind regards,

Laila Adel Ziko

Academic Editor

PLOS ONE
---

## [Editor Report · Acceptance letter]

29 Apr 2024

PONE-D-23-28438R2 

PLOS ONE

Dear Dr. Kadry, 

I'm pleased to inform you that your manuscript has been deemed suitable for publication in PLOS ONE. Congratulations! Your manuscript is now being handed over to our production team.

Kind regards, 

on behalf of

Dr. Laila Adel Ziko 

Academic Editor

PLOS ONE